# Potential of Induced Pluripotent Stem Cells for Use in Gene Therapy: History, Molecular Bases, and Medical Perspectives

**DOI:** 10.3390/biom11050699

**Published:** 2021-05-07

**Authors:** Agnieszka Fus-Kujawa, Barbara Mendrek, Anna Trybus, Karolina Bajdak-Rusinek, Karolina L. Stepien, Aleksander L. Sieron

**Affiliations:** 1Department of Molecular Biology, Faculty of Medical Sciences in Katowice, Medical University of Silesia, Katowice, Medykow 18 Street, 40-752 Katowice, Poland; afus@sum.edu.pl (A.F.-K.); s76362@365.sum.edu.pl (A.T.); kbugaj@sum.edu.pl (K.L.S.); 2Centre of Polymer and Carbon Materials, Polish Academy of Sciences, M. Curie-Sklodowskiej 34, 41-819 Zabrze, Poland; bmendrek@cmpw-pan.edu.pl; 3Department of Medical Genetics, Faculty of Medical Sciences in Katowice, Medical University of Silesia, Katowice, Medykow 18 Street, 40-752 Katowice, Poland; kbajdak-rusinek@sum.edu.pl

**Keywords:** iPSCs, stem cells, gene therapy, differentiation, auto-transplantation, regenerative medicine

## Abstract

Induced pluripotent stem cells (iPSCs) are defined as reprogrammed somatic cells exhibiting embryonic stem cell characteristics. Since their discovery in 2006, efforts have been made to utilize iPSCs in clinical settings. One of the promising fields of medicine, in which genetically patient-specific stem cells may prove themselves useful, is gene therapy. iPSCs technology holds potential in both creating models of genetic diseases and delivering therapeutic agents into the organism via auto-transplants, which reduces the risk of rejection compared to allotransplants. However, in order to safely administer genetically corrected stem cells into patients’ tissues, efforts must be made to establish stably pluripotent stem cells and reduce the risk of insertional tumorigenesis. In order to achieve this, optimal reprogramming factors and vectors must be considered. Therefore, in this review, the molecular bases of reprogramming safe iPSCs for clinical applications and recent attempts to translate iPSCs technology into the clinical setting are discussed.

## 1. Introduction

Since the advent of the term ‘stem cells’ in the 19th century [1], the idea of single-cell capacity to differentiate into any type of tissue has been appealing to both biologists and clinicians. Currently, stem cells are defined as undifferentiated cells capable of self-renewal (extensive proliferation) and with potential to give rise to different cell types [2], and they are found in multicellular organisms in all stages of life. However, adult stem cell renewal is limited because they can only differentiate into cells specific to one tissue, although attempts have been made to prove otherwise [3]. Hematopoietic stem cells in the bone marrow, for instance, are able to give rise only to blood cells. Despite this and other drawbacks, human bone marrow stem cells were the first ones to be used in clinical practice back in 1957 [4] as bone marrow transplantation—now used frequently in the treatment of blood disorders [5]. The more recent focus of stem cell research has broadened their potential in regenerative and personalized medicine and modelling of human diseases; however, finding stem cells capable of differentiating into all types of tissue is challenging. Embryonic stem cells meet these conditions, but their obtaining process is met with opposition of ethical nature [6]. A possible way of overcoming this obstacle has been found in 2006 when Takahashi and Yamanaka published the results of their elegant work on reprogramming mouse embryonic fibroblasts into a new type of cell, called induced pluripotent stem (iPSCs) cells. The iPSCs exhibited morphology, growth properties, and cell marker gene expression of embryonic stem cells [7]. This procedure makes it possible to gather genetically patient-specific stem cells from many types of easily obtainable somatic cells [8]. Methods of creating safe iPSCs for clinical use are still under development [9]; however, the technology holds enormous potential, especially in creating precise medical techniques. In this review, the molecular bases of cell reprogramming and possible applications of iPSCs in this developing branch of medicine, namely gene therapy, are discussed.

## 2. What Are Stem Cells?

A stem cell is an undifferentiated cell that can divide to produce daughter cells that continue as stem cells (self-renew), and some of them are destined to differentiate to specialized cells. Stem cells are an ongoing source of differentiated cells that make up the tissues and organs of animals and plants. Therefore, the medical definition of a stem cell is an unspecialized cell capable of perpetuating itself through cell division and having the potential to give rise to differentiated cells with specialized functions.

### 2.1. Where Do the Stem Cells Originate From?

In nature, stem cells can be isolated from a variety of sources; however, they express a different range of stemness, depending on their origin. Therefore, their classifications also vary. The first criterion for discussion of stem cell classification is based on their origin (Figure 1). Pluripotent stem cells obtained under specific conditions from the inner cell mass (ICM) of an embryo, called embryonic stem cells (ESCs), were first isolated in 1981 [10] and are being currently used only for research. Isolation of the ICM, however, results in the blastocyst’s destruction, which raises serious resistance of the ethical nature.

Another potentially applicable type of stem cells are adult stem cells (ASCs), many of which have roles in postnatal tissue remodeling and repair. Acquiring ASCs does not raise ethical concerns, but as they do not exhibit pluripotency, their applications are limited. However, some studies suggest the expression of pluripotency markers in ASCs, but most of these experiments involved long in vitro cultures or did not demonstrate pluripotency at the single-cell level [3,11].

In 2006, a new type of stem cell was reported by Yamanaka and colleagues [7], who generated iPSCs directly from murine somatic cells. This procedure requires transducing cells with transcription factors, shown previously to be upregulated in tumors or to have a function in maintaining pluripotency in embryos or ESCs. The original set of factors used by Yamanaka’s group included Oct3/4 (octamer-binding transcription factor 3/4), Sox2 (SRY-related high-mobility group box protein-2), oncoprotein c-Myc, and Klf4 (Kruppel-like factor 4). Just a year later, an analogous generation of human iPSCs (hiPSCs) from dermal fibroblasts was described [12]. Yamanaka’s research team used the same reprogramming factors and managed to demonstrate hiPSCs similarity to ESCs in terms of their morphology, presence of surface antigens, gene expression profile, and ability to form in vitro cells of three germ layers. This approach circumvents the ethical concerns regarding ESCs acquisition while delivering fully pluripotent cells. Additionally, since 2006, the cell reprogramming methods have advanced. Furthermore, in this review, several techniques of iPSCs production that are currently used in the laboratories will be presented.

### 2.2. Classification of Stem Cells Based on Their Stemness

Further classification of stem cells appreciates their differentiation potency (Figure 1). As a new organism develops, most of its cells gradually lose their potential and become more specialized. The zygote provides embryo cleavage cells with the highest differentiation potential. This type of stem cell is described as totipotent because it is capable to form both embryo and extraembryonic structures, such as the placenta. About 4 days after fertilization, the embryo transforms into the blastocyst stage, in which a group of cells, called inner cell mass (ICM), become pluripotent. Pluripotent stem cells lose the ability to differentiate into extraembryonic structures; however, they may form cells of all three germ layers, namely ectoderm, mesoderm, and endoderm, from which stems every tissue and organ upon further development. As this process continues, the number and potency of stem cells in the organism decrease. In most adult tissues, multipotent cells can be found. This type of cell differentiates into single germ layer-derived tissues, for example, mesenchymal stem cells (MSCs) give rise to mesoderm-derived tissues, such as muscle, bone, cartilage, or adipose tissue. While differentiating, multipotent cells gradually become oligopotent, as their differentiating potential reduces to several cell lineages (hematopoietic stem cells, e.g., form myeloid or lymphoid cell lines). Eventually, the last type of stem cell arises in the process of specialization to unipotent cells, capable to give rise to only one specific cell type. Unipotent cells still have the ability to self-renew, contrary to progenitor cells, which can only transform to one type of cell [13]. The characteristics of stem cells are presented in Table 1.

## 3. Molecular Mechanisms of Somatic Cells Reprogramming Into iPSCs

The reprogramming of differentiated cells, known also as transformation to early-stage cells, has been known since methods for classical cytogenetics were developed. In terms of transforming differentiated white blood cells, lymphocytes from dividing lymphoblast cells, a chemical such as phytohemagglutinin is added to the cell culture medium. However, this approach allows the cells to be only one step back to the precursor cell stage. Reprogramming differentiated cells a few steps back to the pluripotent stage therefore requires delving deeper and reenabling the silenced genetic programs that are turned on in the pluripotent cells. Observations of native stem cells provided some clues about gene expression profiles and protein markers specific for pluripotent stem cells [14]. This knowledge has been applied for successful reprograming of somatic cells to iPSCs.

### 3.1. Molecular Bases of Cell Reprogramming with Yamanaka and Thomson’s Factors

Yamanaka reprogrammed mouse somatic cells into pluripotent stem cells for the first time in 2006. It was an essential milestone for the field of gene therapy. One year later, Yamanaka reprogrammed human skin fibroblasts in the same way. The mouse and human skin fibroblasts were reprogrammed by introduction of four transcription factors: Oct3/4, Sox2, Klf4, and c-Myc (OSKM; Yamanaka factors) using retroviral vectors [7,12]. These factors are necessary for inducing a state of pluripotency, therefore enabling generation of iPSCs. In recent years, new methods have been made in the field of Yamanaka factors delivery to cells. These methods use Sendai virus as a carrier of cDNA encoding Oct3/4, Sox2, Klf4, and c-Myc factors. Sendai virus (SeV) replicates in the form of negative-sense ssRNA directly in the cytoplasm, thus preventing recombination. Other assets of SeV vectors are highly efficient, have a broad spectrum of target cells, and no identified pathogenicity in human cells [15,16]. This method is commercially available and known in the form of the Cytotune-iPS 2.0 Sendai Reprogramming Kit. The method is also very promising, especially in the potential bench to bedside translational studies.

The mechanism of reprogramming induction in somatic cells by four transcription factors—Klf4, Oct3/4, Sox2, and c-Myc—is still not well understood. It has been documented that Oct3/4 codes a transcription factor containing the POU homeodomain. *C-MYC* codes for a nuclear phosphoprotein that controls the progression of the cell cycle [17]. Evidence has been reported that Oct3/4 and Sox2 are key transcriptional factors that inhibit the expression of genes associated with embryonal stem cell differentiation [18]. However, these factors can not bind to their target genes in differentiated cells because of inhibition mechanisms such as DNA methylation and histones modification [12].

Oct3/4 is crucial for somatic cell reprogramming, and its lack in the mixture of reprogramming factors results in failure of iPSCs colony generation. Interestingly, the modulation of Oct3/4 expression causes extremely low efficiency of the reprogramming process. It has been documented that Oct3/4 also plays a key role in activating ESC-specific genes by interaction with Sox2 and Klf-4. Most of these genes are silenced in partially reprogrammed cells [19,20,21]. As it was shown, Oct3/4 protein is required for stemness properties of both murine and primate ESCs; therefore, it is an essential factor for somatic cell reprogramming. Its activity depends on its posttranslational modifications [22,23]. In general, the activity of a protein, its interactions, localization, and stability may be controlled by posttranslational modifications as well. This novel approach for somatic cell reprogramming allows to eliminate the use of viruses as carriers in experiments of reprogramming. It has been demonstrated that ubiquitination promotes protein degradation and in the case of Oct3/4 leads to regulation of cellular proliferation and differentiation. Taking it into account, it may be concluded that this factor is not only responsible for cell proliferation but also for the control of their differentiation [17].

The second factor that is essential for somatic cell reprogramming is Sox2. It plays an essential role in reversing the epigenetic configuration of differentiated cells back to a pluripotent embryonic state. Sox2 is also critical for directing the differentiation of iPSCs to neural progenitors and for maintaining the properties of neural progenitor stem cells [24]. The levels at which Sox2 and Oct3/4 are expressed during reprogramming of somatic cells to iPSCs are critical, as reported in several studies [18,25,26], and their ratio affects the reprogramming efficiency and quality of iPSCs colonies. It has been demonstrated that the increase in Oct3/4 levels and maintained levels of Sox2, Klf4, and c-Myc, respectively, slightly increased the frequency of reprogramming. On the other hand, the decrease in Oct3/4 and the increase in Sox2 levels resulted in a vivid decrease in the frequency of reprogramming. Interestingly, the decrease in Sox2 levels increased the frequency of partially reprogrammed iPSCs production [26]. Moreover, it has been demonstrated that the increase in Oct3/4 [27] and decrease i Sox2 substantially improves the reprograming of somatic cells to iPSCs that are able to create all forms of iPSCs in mice after injection into tetraploid blastocysts [28]. Furthermore, it has been confirmed that knock-out of Oct3/4 or NANOG in embryonic germ cells resulted in their apoptosis [25,26,29,30]. It may be that the cause of it is the role of these factors in the regulation of cell survival along with suppression of differentiation.

Importantly, it was confirmed that ectopic expression of c-Myc causes cells transformation to tumors in offspring and that retroviruses themselves can cause insertional mutagenesis. The generation of iPSCs with a minimal number of factors may hasten the clinical application of this approach. Therefore, exclusion of c-Myc from the reprogramming mixture is crucial to obtain normal and safe iPSCs [16].

Other factors in the reprogramming of iPSCs are Lin28A and Lin28B. They seem to be important for growth, metabolism, and tissue development of an organism as well as somatic cell reprogramming [19,23,31,32]. Lin28B is induced by factor c-Myc and is responsible for cellular Myc-dependent proliferation. Lin28A is an RNA-binding protein; thus, it is possible that maternal Lin28A might also be involved in ribosomal RNA processing in the nucleolus for the regulation of zygotic genome activation during the maternal-to-zygotic transition (MZT). The earliest phases of embryogenesis feature high levels of Lin28A because of protein inheritance through the maternal oocyte. In mouse zygote to blastocyst preimplantation, Lin28A is localized only in the nucleolus where it is responsible for the regulation of nucleolar maturation [21]. Knockdown of Lin28A in the zygote produced defects in nucleolar morphology and developmental arrest at the two-cell and four-cell stages, suggesting that Lin28A is required for proper nucleolar genesis and function as well as early embryogenesis. In turn, in murine ESCs, Lin28A is localized in the nucleolus, but this is not the case in primate ESCs [33].

Surprisingly, overexpression of Lin28A with pluripotency-associated transcription factors Oct3/4, Sox2, and NANOG helps the promotion of the reprogramming of human somatic fibroblasts into indefinitely self-renewing iPSCs [25,34,35]. These data indicate that Lin28A is critical for pluripotent stem cell self-renewal but does not appear to be essential for pluripotency in vivo [31]. Indeed, it has been demonstrated that Lin28A can affect the early stochastic phase of iPSCs generation by accelerating the cell cycle and that Lin28A is one of the earliest markers of the deterministic phase of somatic reprogramming of iPSCs after endogenous *Sox2* expression is induced [36]. Therefore, the lack of Lin28A or NANOG in the mixture of reprogramming factors causes the decrease in the colony number of iPSCs [18,35,37]. Factors Oct3/4, Sox2, Lin28A, and NANOG are commonly used for somatic cells reprogramming and are called Thomson’s factors.

Ways of somatic cells reprogramming and their differentiation to three germ layers with defined factors are presented in Figure 2.

### 3.2. Vectors in iPSCs Reprogramming

Gamma-retroviral vectors, such as pMXs [12] or pMSCV [38], exhibit high transduction efficiency; however, their potential is limited to proliferating cells. Alternative integrating approaches include lentiviruses with higher transduction efficiency and broader target spectrum [39], or nonviral methods, such as electroporation, liposomes, polycistronic vectors with induction factors driven by one promoter. Probably the most important drawback of incorporating vectors when considering potential clinical applications is insertional mutagenesis. Differentiated cells derived from iPSCs may also exhibit unwanted reactivation of transgenes. The resulting overexpression of proto-oncogenes, such as c-Myc, may lead to tumorigenesis [40].

Another group of vectors circumvents this problem by cutting transgenes out after reprogramming. Some excisable integrating vectors rely on heterologous recombination induced by recombinase (e.g., Cre recombinase) expressed in transduced cells [41]. This enzyme, however, makes acquired cells vulnerable to nonspecific recombination and genomic instability. Another excisable method, based on transposons, mobile genetic elements [42], enables the creation of iPSCs without genetic addition, but it carries similar problems because transposition may occur off-target, affecting the cell’s genomic stability [40].

Possible ways to avoid insertional mutagenesis, apart from removing transgenes, may be not incorporating them in the first place. Non-integrating vectors include episomes and viral vectors. Transient episomal delivery appears promising as viral particles are not needed [43]; however, the efficiency is currently quite low. Integration-defective viral vectors, such as retro, lenti-, or adenoviral, in theory do not fuse with the host’s genome, but in practice random recombination at low rates might occur [44].

Attempts have been made to reprogram cells completely transgene-free by mRNA [45] or protein delivery [46]. These methods seem to be valuable for the future; however, the costs are quite high. RNA delivery requires repeated transfection, which is a challenging process for fragile cell types. In the case of somatic cell reprogramming using proteins that are directly introduced into the treated cells, the efficiency of the process is really low. Importantly, the biggest problem is the plasma membrane that acts as a barrier for the transported proteins because of their size [47].

## 4. Perspectives for iPSCs Use in Disease Therapies

Gene therapy is a rapidly advancing field of medicine, defined as administering nucleic acids to treat, cure, or prevent disorders [48]. These effects can be achieved in various ways, including substitution of the deficient gene or reducing the activity of its harmful products. The development of such methods requires a deep understanding of the disorder’s pathogenesis to create the drug, and running many tests on accurate models to better study its effects before setting up a clinical trial. Animal models have traditionally been used most widely; however, iPSCs hold a great potential as sources of cells imitating diseased ones with acceptable fidelity. Not only do they provide an almost unlimited number of cells genetically identical to the patient’s, but they also enable researchers to obtain copies of difficult-to-access tissues. iPSCs technology is also less expensive compared to animal models. The technology, aside from providing models to test gene therapy on, might prove itself useful in delivering therapeutic agents into the organism. iPSCs derived from patients can be genetically corrected and transplanted back to reconstruct a particular cell line or provide the protein needed. This method, autotransplant, is a promising one, as it surpasses the problem of transplant incompatibility and enables to obtain any type of cell suitable for transplant. Thus, this type of cell modelling has been used in proof-of-concept and preclinical studies of gene therapy for many different diseases. Hence, a few examples of life-threatening genetic disorders are presented in which iPSCs were used as a treatment method (Table 2).

### 4.1. Sandhoff Ddisease

Sandhoff disease is a lysosomal storage disorder characterized by GM2 ganglioside accumulation caused by β-hexosaminidase A and B deficiency. Ganglioside and other metabolite build-up leads to neurodegeneration and early death [63]; however, the direct impact of the disease on early brain development still needs to be understood, mostly due to fetal tissue inaccessibility. Therefore, a study analyzing this linkage with the use of iPSCs was performed [62]. Sandhoff disease iPSCs with mutated β-hexosaminidase β subunit gene (HEXB) and control isogenic corrected iPSCs were generated, from which cerebral organoids, meaning three-dimensional cell culture models, were created. This procedure allowed to examine the level of GM2 storage, which was found already at week 4 of culture in uncorrected organoids in contrast to HEXB-corrected cells, which accumulated considerably less gangliosides. Some genes in corrected organoids were also found to be upregulated relative to disease models; the top nine encoded neuron morphogenesis and central nervous system development factors, indicating more advanced neuronal differentiation.

Therefore, organoids may serve as models for pathophysiology research and gene therapy validation in the future; however, more studies are needed to ensure their fidelity to in vivo structures [64].

### 4.2. SCID—Severe Combined Immunodeficiency

Radiosensitive severe combined immunodeficiency is another disease that could potentially be a target for gene therapy. SCIDs are a group of disorders caused by mutations in genes responsible for the proper function of immune cells, mostly T-, but also B-lymphocytes. Some SCID-causing mutations affect non-homologous end-joining (NHEJ), a DNA repair process, which is essential in T- and B-cell receptor recombination [65], providing TCR and BCR diversity needed for correct lymphocyte function. Apart from T-cell impairment, defective NHEJ may cause somatic cells’ sensitivity to radiation, which in such a case is called RS-SCID. Patients with SCID require lifelong prevention of infections and/or blood forming stem cell transplant [66]. Thus, gene therapy could potentially be an alternative treatment. Developing the therapy strategy, however, requires a model of T-cell lymphopoiesis to evaluate the method’s effectiveness. In a 2015 study, researchers managed to recapitulate T-cell differentiation from iPSCs in vitro with an improved protocol from previous studies [57]. This model allowed them to validate their strategy of gene correction in RS-SCID iPSCs. Cells derived from corrected iPSCs differentiated to CD4+/CD8+ lymphocytes, while uncorrected progenitors stopped at a double-negative thymocyte stage. Another study [67] aimed at creating iPSCs capable of generating mature NK and T cell precursors as a proof-of-concept for the future development of iPSCs-based cell therapy. iPSCs and iPSC-based models of complex biological processes are potentially powerful tools for studying pathomechanisms and creating gene therapies for rare diseases, such as RS-SCID.

### 4.3. Severe Congenital Neutropenia

The precise pathophysiology of severe congenital neutropenia (SCN) has also remained unclear for a long period of time, despite earlier identification of *HAX1* gene mutations as possibly linked to the symptoms. SCN is a myelopoietic disorder that manifests with recurring infections caused by mature neutrophils shortage [68]. SCN patients are treated with granulocyte colony-stimulating factor; however, this therapy carries a risk of severe adverse effects. Therefore, a study was conducted in 2013 to find the connection between *HAX1* mutation and pathological symptoms, and to investigate potential therapies [55]. A neutrophil differentiation system was established from a SCN patient’s iPSCs line. Model myeloid cells at different stages of development were obtained as a valuable resource to perform tests, which would not have been possible to establish using animal models or cultures of differentiated somatic cells. Analyses showed arrest at the myeloid progenitor stage and predisposition to apoptosis, both of which correlated with abnormal granulopoiesis. Moreover, a lentiviral transduction of HAX1 encoding cDNA into iPSCs was conducted, which reversed the disease-related phenotype of the cells (abnormal granulopoiesis and apoptotic predisposition). Thus, iPSCs may serve as a unique research tool for the development of gene therapy of diseases associated with cell differentiation abnormalities.

### 4.4. Hemophilia A

Another health problem, which could be treated with iPSC gene therapy is a debilitating disease, hemophilia A. It is a congenital bleeding disorder caused by dysfunction or, more often, quantitative deficiency of procoagulation factor VIII (FVIII). It results in spontaneous hemorrhaging that can be life threatening, especially when bleeding occurs in the digestive tract, muscles, or brain. Patients require a continuous supply of insufficient protein, usually by intravenous injection of thrombocytes concentrated 3 to 4 times per week. Alternative treatments are sought to improve the patient’s quality of life. Thus, hemophilia A is a potential target for gene therapy. Some clinical trials have already shown advances in this field [69,70].

In 2014, Hideto, Matsui, and coworkers used hiPSCs to test the efficiency of a nonviral delivery system—piggyBac DNA transposon, carrying full-length FVIII cDNA [50]. The expression of the transgene in genetically modified cells was confirmed. Furthermore, mouse models of hemophilia A were transfected, which resulted in endogenous production of FVIII for more than 300 days without developing antibodies to FVIII, which is a major adverse effect of FVIII injection [71]. This study shows that iPSCs may serve as valuable models before proceeding to clinical studies.

Another set of studies [51,52] aimed at obtaining functional endothelial cells (ECs) secreting FVIII. After transducing iPSCs [51], or subsequently ECs [52] with lentiviral vectors, ECs were transplanted into immune-deficient mice. In both cases, cells engrafted and maintained a stable expression of FVIII, which is a promising result for the future development of cell therapy, both for adults and newborns [52].

### 4.5. Osteogenesis Imperfecta

One example of debilitating disease in which patients would benefit from iPSCs gene therapy is congenital osteogenesis imperfecta (OI), known as fragile bone of brittle bone disorder. It is a disease caused by mutations in collagen encoding genes, which makes bones more prone to fragility and brittleness. In 95% of cases, OI is caused by mutations in the *COL1A1* or *COL1A2* genes that encode the alpha1 and alpha2 procollagen type I chains [15,34,72,73]. In clinical practice, the most commonly used criterion is the division of OI into four types according to Sillence classification [74]. Structural defects in collagen may cause changes in its quantity caused by intracellular degradation and/or its worse quality, described as collagen suicide. It is difficult to predict the location of the mutation based on the clinical characteristics of patients with OI. The *COL1A1* gene regions code the amino acids of periods D3 and D4, which play an important role in stabilization of the collagen triple helix. Mutations known to date reduce the stability of this protein. A mutation in the D1 period that codes the sequence results in a clinically milder OI type, but the C-terminus collagen defect causes more severe or even lethal OI types [15,73]. The characteristic symptoms include lower bone mass and fragility, and the appearance of patients includes short and inclined arms, shortened legs, as well as hips turned outwards. In addition, the presence of blue-gray sclera of the eye and abnormalities in the development of dentition known as dentinogenesis imperfecta are characteristic [72,74].

Currently available treatment options of OI include prevention of bone fractures, control of symptoms, and increase in bone mass. The treatment modes include nonsurgical and surgical procedures. Non-surgical approaches including physical therapy, braces, and splints are being used to prevent deformity and promote support and protection, as well as the use of medications. Surgical intervention may be used to deal with local pathologies, such as bone fractures, bowing of bones, or scoliosis. To counteract the overall systemic effects, medications have to be prescribed [15]. Therefore, to improve the therapeutic effects of OI and other genetic diseases of connective tissue caused by mutations in collagen, further investigations are needed on experimental approaches, in particular based on stem cell transplantation and genetic engineering, which at present constitute an interesting and well-developed niche of studies [34,75,76].

### 4.6. Myotonic Dystrophy

Patients with myotonic dystrophy type 1 (DM1) suffer a variety of life-threatening symptoms, mainly myopathy, myotonia, and cardiac conduction defects [77]. DM1 and other repeat expansion dominant diseases, however, are presently a special challenge for gene therapy development, as adding a functional copy of the mutated gene is not sufficient to reverse the pathology. Fixing DM1′s underlying cause, expanded CTG-trinucleotide repeats in the myotonic dystrophy protein kinase (*DMPK*) gene, requires selective excision, which is a challenging procedure [77]. A system of clustered, regularly interspaced, short palindromic repeats (CRISPR/Cas9) holds potential for these types of corrections. A 2018 proof-of-concept study aimed at assessing CRISPR/Cas9 efficiency in DM1 [60]. Patient myoblasts and fibroblasts were reprogrammed into iPSCs, which served as a source of myogenic cell models of the most severely affected tissue. After gene editing, iPSCs’ derived myogenic cells allowed not only for performing a rapid triplet repeat primed PCR (TP-PCR) assay to validate the method’s efficiency, but also for assessment of its effects on the physiology of the cell. Before treatment, iPSCs and their derivatives accumulated ribonuclear foci (RF), one of the most prominent characteristics of DM1. After editing, cells exhibited a significant decrease in RF in the nucleus, consistent with the excision of the CTG repeats. Apart from modelling pathophysiology, corrected myogenic cells may potentially be used to replace the dysfunctional tissue in the future [61]. In another study [61], neural stem cells were derived from human DM1 iPSCs. A correction by integrating an editing cassette containing polyA signals upstream of the CTG repeats led to phenotype reversal, which is a promising step towards autologous iPSCs transplant development in the future.

Despite that patient-specific iPSCs are a valuable material for disease modeling, their use for clinical purposes is expensive. It is valuable to use a limited number of iPSCs lines in order to treat a large number of patients in clinical conditions. A great perspective for these cells is use of human leukocyte antigen (HLA)-Homozygous iPSCs from accessible and less invasive tissues, such as blood [78]. Expression of HLA proteins on the cell surface allows to distinguish self and foreign cells by the immune system [67]. For the first time, iPSCs were generated from peripheral blood cells in 2009, using retroviral vectors for Yamanaka factors delivery. iPSCs derived from blood are ESC-like with respect to morphology, surface antigens expression, and the ability to differentiate in vitro [79]. iPSCs generation was also reported for CD34+ cord blood cells and peripheral blood mononuclear cells (PMNCs) using episomal vectors [78,80]. Such cells may be also generated using non-integrating Sendai virus encoded reprogramming Yamanaka factors [81,82]. The lack of incorporation of the Sendai virus into the host’s genome allows safe use of iPSC cells in clinical practice. Importantly, using of blood cells in order to generate iPSCs allows to reduce invasiveness of the whole procedure. Blood collection is a standard test in clinical practice, and HLA-matching allows to limit alloimmune responses.

### 4.7. Retinitis Pigmentosa

Autosomal recessive retinitis pigmentosa (RP), one of the most common forms of inherited retinal degeneration, is caused mostly by defects in Membrane Frizzled-Related Protein (MFRP) [83]. It is characterized by the disruption and loss of cells in the retina. In the 2014 reported study results, patient-specific iPSCs were established to examine the pathophysiology of retinal pigment epithelium (RPE) and test a gene therapy approach [53]. In MFRP-deficient iPSCs derived from two RP patients, it was possible to identify the destabilization of cells’ actin organization, which suggests a role of this gene in cytoskeleton coordination. In terms of gene therapy testing, AAV8 vector expressing human MFRP was used in both iPSCs and murine preclinical models. Deficient iPSCs were rescued by means of pigmentation and transepithelial resistance recovery, alongside mice demonstrating a favorable response. Patient-specific cell lines might be complementary to mouse models in preclinical studies as this combination reduces the problem of differences in phenotypic expression between human and model animal organisms.

In another study [54], human iPSCs-derived retinal pigmented epithelium cells were transplanted into RP mouse models. Mice did not develop tumors and demonstrated improved visual function over time, which is a promising result for future research on iPSCs autografts in RP.

## 5. Conclusions

The field of gene therapy is constantly advancing. Much effort is dedicated to make the cells completely safe for patients and to minimize the risk of grafted cell rejection, or tumorigenesis. The emphasis should be on the efficiency of the reprogramming process. Reprogramming factors introduced in the form of proteins into somatic cells should be carefully selected, and the carrier should be precisely assessed in terms of safety. Somatic cells reprogramming is a promising method of iPSCs generation. Appropriate selection of reprogramming factors enables an effective and safe process. Unlike ESCs, iPSCs do not arouse ethical controversy, which allows them to be safely used for gene therapy purposes. Taking into account the ability of iPSCs to differentiate into three germ layers, they are a valuable source of other cell type precursors. More in-depth studies are needed to test the potential of iPSCs for gene therapy for many diseases to develop a new treatment strategy that uses the patient’s cells.

## Figures and Tables

**Figure 1 biomolecules-11-00699-f001:**
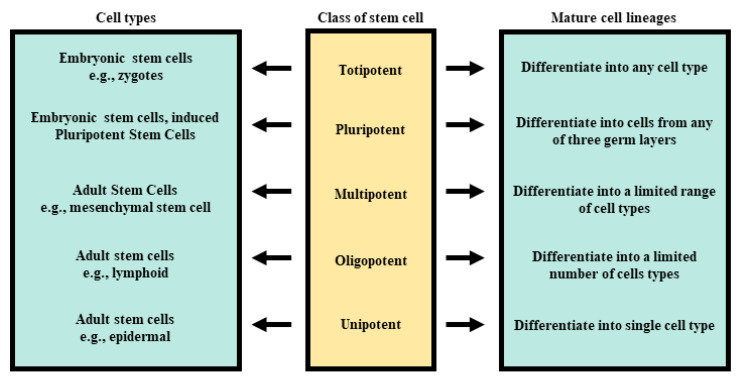
The potential of stem cell differentiation depends on the source of their origin.

**Figure 2 biomolecules-11-00699-f002:**
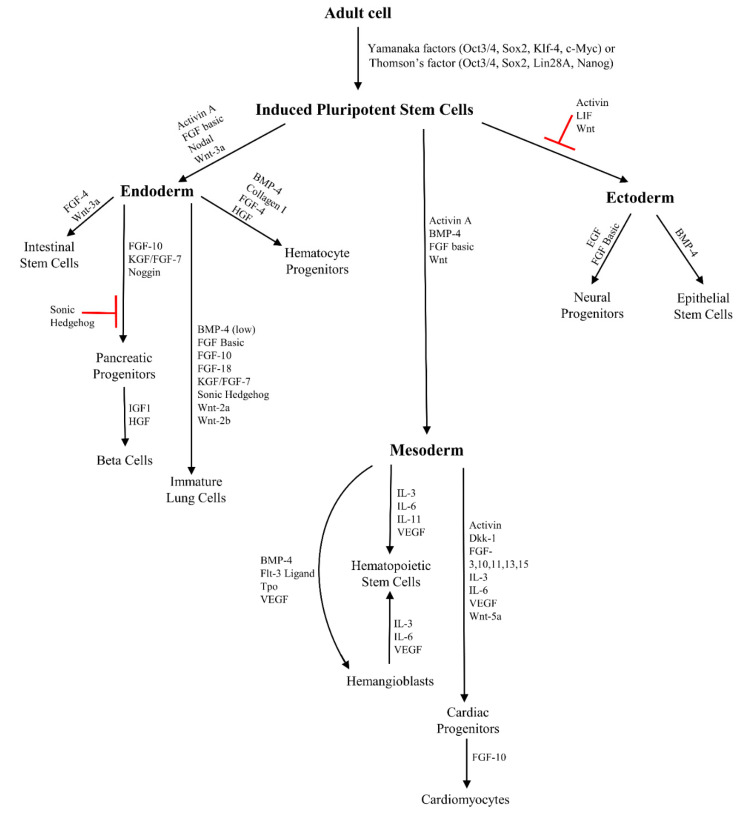
Schematic of somatic cells reprogramming with Yamanaka (Oct3/4, Sox2, Klf-4, and c-Myc) or Thomson (Oct3/4, Sox2, Lin28A, and NANOG) and their potency to differentiation with lineage-specific markers.

**Table 1 biomolecules-11-00699-t001:** Summary of characteristics considered in stem cell classification.

Stem Cell Origin	Stem Cell Type	Potential to Differentiation
Embryonic germ stem cells, e.g., zygotes	Totipotent	3 germ layers and their derivatives and extraembryonic tissue
Embryonic stem cells	Pluripotent	3 germ layers and their derivatives
iPSCs		Trophectoderm
Neural stem cells	Multipotent	Glial and neuronal cells
Stem cells in bone marrow, hematopoietic stem cells	Multipotent	All blood cell types
Mesenchymal stem cells	Multipotent	cells of the mesenchymal lineage (adipocytes, osteocytes, and chondrocytes)
Adult stem cells, e.g., epithelial stem cells, skin stem cells	Multipotent	e.g., stem cells within the bulge, intestinal epithelium
	Unipotent	e.g., the interfollicular epidermis, sebaceous glands, intestinal epithelium
Adult stem cells, e.g., myeloid	Oligopotent	Granulocytes, monocytes, platelets
Adult stem cells, e.g., lymphoid		Lymphocytes, natural killer cells
Spermatogenic stem cells	Unipotent	Sperm cells

**Table 2 biomolecules-11-00699-t002:** Application of induced pluripotent stem cells in gene therapy.

Disease (OMIM Number)	Affected Genes/Proteins	Clinical Features/ Symptoms	iPSCs Application in In Vitro Models	iPSCs Have Potential Applications in Autologous Transplants
Osteogenesis imperfecta (#166200)	COL1A1/COL1A2	lower bone mass and fragility; Short and inclined arms; Shortened legs; Blue-gray sclera of the eye	[49]	-
Haemophilia A (#306700)	Procoagulation factor VIII	Spontaneous haemorrhages	[50]	[51,52]
Retinitis pigmentosa (#500004)	Frizzled-Related Protein (MFRP)	The disruption and loss pf cells in the retina	[53]	[54]
Severe congenital neutropenia (#618752)	HAX1	recurring infections	[55,56]	-
SCID (#602450)	Genes responsible for the proper function of immune cells (mostly T- lymphocytes, but also B-lymphocytes)	T-cell impairment; somatic cells’ sensitivity to radiation;	[57]	[58,59]
Myotonic Dystrophy (#602668)	DMPK1	Myopathy; myotonia; cardiac conduction defects;	[60]	[61]
Sandhoff disease (#268800)	β-hexosaminidase A and B deficiency	Neurodegeneration;	[62]	-

## Data Availability

No new data were created or analyzed in this study. Data sharing is not applicable to this article.

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
