# Peer review of "Potential of Induced Pluripotent Stem Cells for Use in Gene Therapy: History, Molecular Bases, and Medical Perspectives"

_biomolecules, 2021, doi:10.3390/biom11050699_

Round 1

Reviewer 1 Report

In the present manuscript, Fus-Kujawa and co-authors described the remarkable characteristics of induced pluripotent stem cells. The authors here delineated the properties of induced pluripotent stem cells and present a summarize of the possible uses of iPSCs in the developing branch of medicine, gene therapy.

In order to achieve the merit of publication, I would suggest that authors provide a major revision and argue the following points:

Major comments:

1) Please, clarify in section 3, paragraph 3.1, the sentences about Yamanaka’s reprogramming method because it has been mentioned a reprogramming strategy based on the Sendai virus, whereas Yamanaka introduced the 24 candidate genes by retroviral transduction.

2) Please, check figure 2 because it seems to be wrongly cut and add more details to the part related to the ways of somatic cell reprogramming because it is not enough described (e.g., display the Yamanaka's or Thomson's factors).

Minor comments:

Please check the text because some typos are present

Author Response

Katowice, April 27th 2021

Dear Reviewer,

Please find enclosed the revised version of the manuscript biomolecules-1196946 entitled “Potential of induced pluripotent stem cells for use in gene therapy: history, molecular bases, and medical perspectives” submitted to “Biomolecules” as a Review Article.

The manuscript has been revised and rewritten according to the reviewers´ comments (Please see below the point-by-point responses to all raised questions). The main changes made on the manuscript are highlighted.

            We do hope that all amendments needed to provide a clear report have been considered, and that the paper is acceptable for the publication in Biomolecules.

Yours sincerely,

Aleksander L. Sieron

Enclosure: Point-by-point response to the reviewers´ comments.

Point-by-point response to the comments.

Reviewer 1

General Comment: In the present manuscript, Fus-Kujawa and co-authors described the remarkable characteristics of induced pluripotent stem cells. The authors here delineated the properties of induced pluripotent stem cells and present a summarize of the possible uses of iPSCs in the developing branch of medicine, gene therapy.

In order to achieve the merit of publication, I would suggest that authors provide a major revision and argue the following points:

Response to General Comment: We very much appreciate the overall positive evaluation of our review. We are grateful for valuable comments.

Major comments:

Comment 1: Please, clarify in section 3, paragraph 3.1, the sentences about Yamanaka’s reprogramming method because it has been mentioned a reprogramming strategy based on the Sendai virus, whereas Yamanaka introduced the 24 candidate genes by retroviral transduction.

Response to Major Comment 1: Section 3.1 has been corrected according to reviewer’s comment. Somatic cells reprogramming with Yamanaka factors has been clarified as follow “One year later, Yamanaka has reprogrammed human skin fibroblasts in the same way. The mouse and human skin fibroblasts were reprogrammed by introduction of four transcription factors: Oct3/4, Sox2, Klf4, and c-Myc (OSKM; Yamanaka factors) using retroviral vectors [7,12]. These factors are necessary for inducing state of pluripotency, therefore enable generation of iPSCs. In recent years, new methods have been made in a field of Yamanaka factors delivery to cells. These methods use Sendai virus as a carrier of cDNA encoding Oct3/4, Sox2, Klf4, and c-Myc factors.”

Comment 2: Please, check figure 2 because it seems to be wrongly cut and add more details to the part related to the ways of somatic cell reprogramming because it is not enough described (e.g., display the Yamanaka's or Thomson's factors).

Response to Major Comment 2: The figure 2 has been uploaded again in correct form.

Minor comments: Please check the text because some typos are present.

Response to minor comments: We have accordingly corrected typos throughout the MS.

Reviewer 2 Report

Dear editor,

The authors reviewed clinical potential of human iPSCs to therapy. Overall, the MS is scientifically sound and logical. There are some minor points to improve.

  1. In the MS, there is no mentioning of MHC (HLA)-Homozygous iPSCs. It is too expensive to use patient specific iPSCs, so it is clinically important to use limited number of iPSC lines to treat a wide range of patients. As the review focuses on the potential in therapy, this part should be included and discussed.
  2. The Oxford comma should be used or not used consistently throughout the MS.
  3. There should be no full stops at the titles.
  4. Human genes should be uppercase (i.e. NANOG) and mouse – not (i.e. Nanog). There are some clear mistakes throughout the MS.
  5. There is strange logic in table 1. A) It seems like “iPSCs” are of embryo origin, which is not the case. B) “Pluripotent “ cells could give rise only to 3 germ layers. What about the ability of human iPSCs to differentiate into trophectoderm? C) “Adult bone marrow” are multipotent. It follows from the table, that there are no more multipotent stem cells. D) “Adult” and “Adult tissues” classifiers do not make sense and in reality do not correspond to “oligopotent” and “unipotent”. The whole table should be remade to be logical.
  6. Lines 128-131. How could “observations” made in 2017 [14] “provide some clues” and “being applied” for reprogramming in 2006?
  7. Line 135-137. “The mouse and human skin fibroblasts were reprogrammed using Sendai virus-based carrier systems for delivery to the cells” Before that sentence the work from 2006 is cited, Yamanaka used retroviral, not Sendai-based vectors.
  8. Line 137. “factor” --- “factors”
  9. Lines 145-146. “difficult elimination of SeV after the reprogramming is completed” Episomes from the cited Sendai-based reprogramming system (ThermoFisher) are lost with every cell division with a rate about 5%. It is known, that this process is not always efficient, or the episomes could rarely be incorporated into the genome, but generally it is not a problem.
  10. Lines 147-148. “Mechanism [one]…are [many]”
  11. Line 149. “C-myc” – incorrect gene name.
  12. Lines 189-191. The fact that retroviruses cause insertional mutagenesis could not by any means be “Striking”.
  13. Line 194. “Another factor… are”.
  14. Line 196. What is “Myc”? C-Myc, N-Myc, C-MYC, N-MYC…
  15. Right part of the figure 2 is missing.
  16. Figure 2. If “Yamanaka or Thomson factors” are used, it means that this figure represents human iPSCs properties; it should be specified in the text/figure capture.
  17. Line 262. “difficult-to--to-access”
  18. Line 294. “SCID- severe” --- “SCID – severe”

Overall, MS could be accepted after minor changes.

Author Response

Katowice, April 27th 2021

Dear Reviewer,

Please find enclosed the revised version of the manuscript biomolecules-1196946 entitled “Potential of induced pluripotent stem cells for use in gene therapy: history, molecular bases, and medical perspectives” submitted to “Biomolecules” as a Review Article.

The manuscript has been revised and rewritten according to the reviewers´ comments (Please see below the point-by-point responses to all raised questions). The main changes made on the manuscript are highlighted.

            We do hope that all amendments needed to provide a clear report have been considered, and that the paper is acceptable for the publication in Biomolecules.

Yours sincerely,

Aleksander L. Sieron

Enclosure: Point-by-point response to the reviewers´ comments.

Point-by-point response to the comments.

Reviewer 2

General Comment: The authors reviewed clinical potential of human iPSCs to therapy. Overall, the MS is scientifically sound and logical. There are some minor points to improve.

Response to General Comment: We very much appreciate the overall positive evaluation of our review.

Comment 1: In the MS, there is no mentioning of MHC (HLA)-Homozygous iPSCs. It is too expensive to use patient specific iPSCs, so it is clinically important to use limited number of iPSC lines to treat a wide range of patients. As the review focuses on the potential in therapy, this part should be included and discussed.

Response to Comment 1: We appreciated the reviewer’s comment. MHC (HLA)-Homozygous iPSCs as an alternative for use of patient’s specific iPSCs is mentioned and explained in the last paragraph of section 4.5 of the revised manuscript as follows. “Despite patient-specific iPSCs are a valuable material for disease modeling, their use for clinical purposes is expensive. It is valuable to use limited number of iPSCs lines in order to treat large number of patients in clinical conditions. A great perspective for these cells is use of human leukocyte antigen (HLA)-Homozygous iPSCs from accessible and less invasive tissues, such as blood [79]. Expression of HLA proteins on the cell surface allows to distinguish self and foreign cells by immune system [80]. For the first time, iPSCs were generated from peripheral blood cells in 2009, using retroviral vectors for Yamanaka factors delivery. iPSCs derived from blood are ESCs-like with respect to morphology, surface antigens expression, and the ability to in vitro differentiation [81]. iPSCs generation was also reported for CD34+ cord blood cells and peripheral blood mononuclear cells (PMNCs) using episomal vectors [79,82]. Such cells may be also generated using non-integrating Sendai virus encoded reprogramming Yamanaka factors [83,84]. The lack of incorporation of the Sendai virus into the host's genome allows safe use of iPSCs cells in clinical practice. Importantly, using of blood cells in order to iPSCs generation allows to reduce invasiveness of whole procedure. Blood collection is a standard test in clinical practice and HLA-matching allows to limit alloimmune responses.”.

Comment 2: The Oxford comma should be used or not used consistently throughout the MS.

Response to Comment 2: We have accordingly updated the Oxford comma throughout the MS.

Comment 3: There should be no full stops at the titles.

Response to Comment 3: We have accordingly updated all titles.

Comment 4: Human genes should be uppercase (i.e. NANOG) and mouse – not (i.e. Nanog). There are some clear mistakes throughout the MS.

Response to Comment 4: We have considered it accordingly, all names of human and mouse genes have been corrected.

Comment 5: There is strange logic in table 1. A) It seems like “iPSCs” are of embryo origin, which is not the case. B) “Pluripotent “ cells could give rise only to 3 germ layers. What about the ability of human iPSCs to differentiate into trophectoderm? C) “Adult bone marrow” are multipotent. It follows from the table, that there are no more multipotent stem cells. D) “Adult” and “Adult tissues” classifiers do not make sense and in reality do not correspond to “oligopotent” and “unipotent”. The whole table should be remade to be logical.

Response to Comment 5: We have considered it accordingly, changes have been introduced to table 1.

  1. A) Origin of pluripotent stem cells has been corrected. This line has been divided in order to clarify this classification.
  2. B) Ability of iPSCs to differentiation to trophectoderm has been added to table.
  3. C) Other examples of multipotent stem cells have been added to table.
  4. D) More specific description of oligopotent and unipotent stem cells has been introduced to table.

Comment 6: Lines 128-131. How could “observations” made in 2017 [14] “provide some clues” and “being applied” for reprogramming in 2006?

Response to Comment 6: We have replaced wrong article at position [14] by the correct one.

Comment 7: Line 135-137. “The mouse and human skin fibroblasts were reprogrammed using Sendai virus-based carrier systems for delivery to the cells” Before that sentence the work from 2006 is cited, Yamanaka used retroviral, not Sendai-based vectors.

Response to Comment 7: Section 3.1 has been corrected accordingly.

Comment 8: Line 137. “factor” --- “factors”

Response to Comment 8: This has been corrected accordingly.

Comment 9: Lines 145-146. “difficult elimination of SeV after the reprogramming is completed” Episomes from the cited Sendai-based reprogramming system (ThermoFisher) are lost with every cell division with a rate about 5%. It is known, that this process is not always efficient, or the episomes could rarely be incorporated into the genome, but generally it is not a problem.

Response to Comment 9: We appreciated the reviewer’s comment. This sentence has been removed from Manuscript.

Comment 10: Lines 147-148. “Mechanism [one]…are [many]”

Response to Comment 10: This has been corrected accordingly.

Comment 11: Line 149. “C-myc” – incorrect gene name.

Response to Comment 11: This has been corrected accordingly.

Comment 12: Lines 189-191. The fact that retroviruses cause insertional mutagenesis could not by any means be “Striking”.

Response to Comment 12: Above-mentioned word has been properly replaced.

Comment 13: Line 194. “Another factor… are”.

Response to Comment 13: This has been corrected accordingly.

Comment 14: Line 196. What is “Myc”? C-Myc, N-Myc, C-MYC, N-MYC…

Response to Comment 14: This has been clarified.

Comment 15: Right part of the figure 2 is missing.

Response to Comment 15: We very much appreciate the overall positive evaluation of our review. The figure has been uploaded again in correct form.

Comment 16: Figure 2. If “Yamanaka or Thomson factors” are used, it means that this figure represents human iPSCs properties; it should be specified in the text/figure capture.

Response to Comment 16: Names of Yamanaka and Thomson’s factors have been put on figure 2 in order to make it more specific and clear.

Comment 17: Line 262. “difficult-to--to-access”

Response to Comment 17: This has been corrected accordingly.

Comment 18: Line 294. “SCID- severe” --- “SCID – severe”

Response to Comment 18: This has been corrected accordingly.

Sincerely

Aleksander L. Sieron

Round 2

Reviewer 1 Report

In the present manuscript, Fus-Kujawa and co-authors described the remarkable characteristics of induced pluripotent stem cells. After the first revision, the results are more solid and well described.